# Investigating Delayed Rupture of Flow Diverter-Treated Giant Aneurysm Using Simulated Fluid–Structure Interactions

**DOI:** 10.3390/bioengineering12030305

**Published:** 2025-03-18

**Authors:** Pablo Jeken-Rico, Yves Chau, Aurèle Goetz, Jacques Sedat, Elie Hachem

**Affiliations:** 1Mines Paris, Université PSL, Centre de Mise en Forme des Matériaux (CEMEF), UMR7635 CNRS, 06904 Sophia Antipolis, France; aurele.goetz@minesparis.psl.eu (A.G.); elie.hachem@minesparis.psl.eu (E.H.); 2Interventional Neuroradiology Department, Nice University Hospital, 06100 Nice, France; chau.hd@chu-nice.fr (Y.C.); sedat.j@chu-nice.fr (J.S.)

**Keywords:** intracranial aneurysms, fluid–structure interaction, delayed rupture, flow diverter

## Abstract

Giant intracranial aneurysms are frequently treated shortly after discovery due to their increased risk of rupture and commonly symptomatic nature. Among available treatments, flow diverters are often the sole viable option, though they carry a rare but serious risk of delayed post-operative rupture. The underlying mechanisms of these ruptures remain unknown, due to the biomechanical complexity of giant aneurysms and challenges in replicating in vivo hemodynamic conditions within numerical simulation frameworks. This study presents a novel fluid–structure interaction simulation of a giant intracranial aneurysm treated with a flow diverter, based on high-resolution rotational angiography imaging. The resulting hemodynamics are compared to three established delayed-rupture hypotheses involving pressure rises, chaotic flow and autolysis. When considering wall compliance, the analysis reveals a consistent phase shift, dampening in pressure cycles, and an increased aneurysmal flow. These findings highlight the need for revisiting existing hypotheses and provide a foundation for advancing both computational modelling and clinical management strategies for giant intracranial aneurysms.

## 1. Introduction

Giant Intracranial Aneurysms (IAs) represent around 5% of all aneurysmal formations [1]. Nevertheless, this subgroup poses a significant burden to patients and clinicians due to their 20-fold higher rupture risk compared to smaller lesions [2,3]. Furthermore, their large size (>25 mm) and commonly ample pulsatility (see Figure 1) often lead to neurological symptoms even in their unruptured state [1]. For these reasons, giant IAs are frequently treated, despite a higher risk of peri- and postprocedural complications [3].

From the available treatment options, many are unsuitable for giant IAs due to their large neck area and their symptomatic volume. Surgical clipping, for instance, is associated with increased morbidity, particularly when the aneurysm fundus is located in the posterior circulation [4]. Endovascular coiling is similarly limited due to insufficient coil support and an increased risk of detachment or coil migration [4]. Flow diverters (FDs) address these challenges through their placement in the parent artery. Once deployed, these low-porosity braided mesh devices reduce intra-aneurysmal flow, promoting stable thrombus formation within the sac and effectively reducing rupture risk. However, deployed FDs necessitate the use of antiplatelet drugs to prevent thromboembolic complications, which can, paradoxically, jeopardize the desired treatment outcomes [5].

Despite their benefits, a rare but severe complication following the deployment of FDs is the delayed rupture of the aneurysmal wall. This phenomenon typically occurs between several days to months post-intervention, disproportionately affecting large and giant IAs [6,7]. The precise mechanism of delayed rupture remains elusive, likely due to the complexity and rarity of these cases. Nonetheless, several hypotheses have been proposed based on clinical observations and autopsy findings. In 2011, Cebral et al. [8] performed numerical simulations on seven patients, three of whom experienced rupture within a week of FD treatment. Their analysis of computed hemodynamics suggested that increased intra-aneurysmal pressure might be a contributing factor to these adverse outcomes. Around the same time, Kulcsár et al. [9] reviewed 13 cases of delayed rupture after FD treatment. Their observations revealed extensive thrombosis in all cases at the time of rupture, leading them to propose thrombus-mediated autolytic cascades as a potential mechanism for aneurysm wall weakening. In 2019, Li et al. [10] advanced this line of inquiry by comparing the computed hemodynamics of 10 patients with delayed rupture against 20 patients with successful treatment outcomes under similar conditions. Their findings highlighted the emergence of unstable flow within the aneurysm bulge as the only statistically significant indicator associated with delayed rupture.

Giant and large IAs pose a particular modelling challenge due to the frequently observed pulsatility of their walls [11]. Recent advances in medical imaging technology progressively reveal these aneurysm pulsations, with the larger lesions being easier to screen given the limited available resolutions [12,13,14]. The periodic bulge volume change can induce a capacitance effect, thereby altering the pressure profiles. At the same time, hemodynamic indicators, such as the Wall Shear Stress (WSS), are affected by the change in flow patterns and incidence angles. Despite this, taking into account these wall movements through Fluid–Structure Interaction (FSI) simulations has remained notoriously rare due to the algorithmic complexity it entails and the lack of knowledge regarding material properties and wall thicknesses of heterogeneous diseased tissue [15,16].

In this study, we examine the case of a patient who suffered a delayed rupture after FD treatment. Unlike previous studies, the availability of high-time-resolution angiographic images allowed us to calibrate the material properties of the aneurysm wall. Using these parameters, we conducted novel simulations incorporating both the embedded FD and fully coupled FSI modelling of the aneurysm bulge to evaluate pre- and postoperative hemodynamics. The results were analyzed in the context of previously proposed hypotheses [8,9,10], highlighting the changes induced through the wall compliance modelling.

## 2. Methods

### 2.1. Patient Record

The 54-year-old patient, with no history of smoking, hypertension, or familial aneurysms experienced vertigo, anxiety, and unusual sensations, caused by the presence of a giant aneurysm in the left supra-clinoid Internal Carotid Artery (ICA). After other treatment options were excluded due to the aneurysm morphology and size, a 3 mm FRED Jr flow-diverter (Terumo Neuro, Aliso Viejo, California) was deployed, immediately causing contrast stasis in the aneurysm sac and a widening of two stenoses located just up- and downstream of the aneurysm neck. Six months after the procedure, a follow-up cerebral arteriography showed a patent stent with unchanged opacification of the aneurysm sac. Dual antiplatelet therapy with aspirin (75 mg/day) and prasugrel (10 mg/day) was continued.

Eight months after the intervention, the patient reported a sudden headache. A Computed Tomography (CT) scan revealed spontaneous hyperdensity in the aneurysm, suggesting partial thrombosis. A week later, the patient suffered a sudden coma, with a Glasgow Coma Scale score of 4 caused by subarachnoid haemorrhage and a left temporal hematoma. Cerebral arteriography showed persistent aneurysm opacification, a patent stent, and active contrast leakage from the posteroinferior part of the aneurysm sac, marked in red on Figure 2. Rotational angiography (6 fps) revealed noticeable wall movement that could not previously be seen (see Figure 1). The coil deployment stopped bleeding and occluded the aneurysm, but did not prevent the patient’s death 5 days later.

### 2.2. Mesh Generation

The computational study is based on the digital subtraction angiography images with a voxel size of 0.274 mm taken before the first intervention and on the day of the rupture. The segmentation was carried out using the open-source 3DSlicer [17] and checked with neurointerventionalists with more than a decade of experience. The surface meshes were extruded along the open boundaries as described in [18] and meshed with isotropic tetrahedra size between 0.25 and 0.3 mm. Unstructured boundary layers with an initial thickness of 20 μm and a growth factor of 1.2 were generated using anisotropic remeshing to improve the resolution in the walls’ proximity.

The postoperative images were virtually stented using the algorithm described in a previous contribution [19]. The embedded stent contains a total of 52 wires with a thickness of 30 μm each, following manufacturer indications. The resulting average porosity amounts to 71.4% at the neck and was confirmed by segmenting the guide wires from the angiography images. The FD was embedded into the computational mesh employing levelset splitting and anisotropic meshing [19]. The resolution around the extracted device was set to 15 μm and was linearly scaled up to the background mesh size over a distance of 0.4 mm. A view of the resulting mesh, featuring 8M tetrahedral elements, is provided in Figure 3.

To save computational effort for the costly FSI simulations, we tested using a reduced FD stent, containing only the portions facing the IA and the branching anterior cerebral artery. Comparing both devices based on the intra-aneurysmal pressure, the velocity magnitude and the Pulsatility Index showed no relevant differences (see Appendix B).

### 2.3. Fluid Dynamics

The fluid dynamics are governed through the transient, incompressible Navier–Stokes equations and were resolved using an in-house finite-element solver [20]. The scheme utilizes a Variational Multiscale-type Stabilization (VMS) to ensure compatibility for same-order interpolations of velocities and pressure. The convection term accounts for a mesh velocity used by the Arbitrary Lagrangian–Eulerian framework in the FSI simulations. The time integration is carried out with a second-order accurate semi-implicit scheme using a fixed timestep size of 1 ms to achieve approximately unit Courant numbers.

The inflow was dictated by a generalized waveform for patients in the same age range [21] that matched the observed cardiac rhythm of the patient. The inflow was set through a centred parabolic profile at the ICA. The outflows were modelled using Murray’s law (n = 3) following the steps described in a previous work [18]. The shear-thinning rheology of blood was approximated using the modified Casson model [22], in which we incorporated the patient-specific hematocrit of 34.1%. The fluid’s density was assumed constant at 1056 kg/m^3^ [23]. The resulting fluid velocities were used to compute the WSS and two further quantities, the Time-Averaged Wall Shear Stress (TAWSS) and the Oscillatory Shear Index (OSI).

### 2.4. Coupled Solid Mechanics

Arterial tissue was modelled as hyperelastic with a Neo-Hookean formulation and a Simo–Taylor volumetric model [24,25]. The solid dynamics were discretized following our updated Lagrangian mixed displacement–pressure VMS formulation. To achieve strong coupling and ensure continuity of displacements and stresses at the solid–fluid interface, a partitioned coupling scheme with under-relaxed fixed-point iterations and Aitken $\Delta^{2}$ acceleration was employed with a $10^{-7}$ m tolerance [26,27]. The fluid mesh was adapted to the solid displacement using a diffusion-based mesh adaptation method [28,29]. Further details on the FSI environment are available in Goetz et al. [16].

The solid modelling was limited to the aneurysm bulge, as no significant movement was observed in the parent artery or its branches. A solid mesh was extruded from the existing fluid mesh, comprising four layers of tetrahedral elements to achieve a total wall thickness of 0.5 mm (see Figure 3). The Young’s modulus was set to 1 MPa throughout most of the aneurysm, with a reduced value of 0.1 MPa at the tip, as shown in Figure 4. This adjustment reflects the significant movement observed in medical imaging at the tip, a region facing the lateral ventricle. Unlike other sections constrained by surrounding brain tissue, this area experiences less restriction, allowing for greater deformation. These chosen properties have been adjusted to visually match the available rotational angiography data.

## 3. Results

### 3.1. Hemodynamics

The flow in the untreated aneurysm is characterized by a large stable vortex which can be seen in the upper row of Figure 5. While this vortex retains shape and direction throughout the cardiac cycle, the ones found in the treated cases undergo a $70^{\circ }$ sweep and feature some secondary recirculation zones. The bulge volume change in the latter reaches 9.2% over a full cardiac cycle, falling in line with literature values [12,13,14].

The TAWSS shown in the second row of Figure 5 reflects a decrease of 10 Pa after deploying the FD stent. The comparison between the rigid and compliant simulations returns only small quantitative differences in the range of 0.5 Pa. The oscillatory character of the shearing, displayed one row below follows a similar trend. The structured flow in the preoperative case shows only localized OSI peaks at the vortex centre and in a pouch on the left side of the IA. The less stable flow in the postoperative cases causes larger recirculation zones, thereby increasing the OSI over large parts of the bulge. The higher inflow of the FSI simulation results in a moderate increase in flow stability and therefore in smaller OSI magnitudes, though the differences remain mostly quantitative.

To better assess the flow in the bulge throughout the cardiac cycle, we consider the bulge-averaged velocity $\bar{u}$ shown in Figure 6. This measure, often used when evaluating the effect of FD, shows here too a reduction in the flow by 59.7% postoperatively, as commonly recorded in the literature [30,31]. Agreeing with the previously observed changes, the FSI case displays a lower flow reduction of 55% with respect to the rigid-wall preoperatory case as a consequence of the inflating and deflating mechanics of the bulge, which induce larger flow rates in the treated configuration.

### 3.2. Pressure

Extending the analysis of velocity and WSS, pressure emerges as not only a key hemodynamic metric but also notably resilient to solver settings, ensuring consistency across numerical algorithms [11,32]. The first factor altering the pressure distributions in the present case is the geometrical constraints of the arterial network. In the preoperative state, the ICA at the distal end of the aneurysm features a diameter of 2.15 mm lying notably below the common sizes recorded by Baz et al. [33]. This stenosis reflects on the pressure along the vessel centerline as seen on the top row of Figure 7, which drops by approximately 2.5 kPa alone at the terminal section during systole. After the stent deployment, the vessel was widened at the terminal sections, removing the flow impediment. At the same time, the aneurysm’s neck has retracted away from the bulge leading to a minor linear decay at this section compared to when the entire ICA flow passed through the IA (see Appendix A for more details). In all cases, the pressure over the bulge was nearly homogeneous, as values remained within 10 Pa of the neck pressure, confirming previous observations in the literature [8,30]. Secondly, the compliance of the aneurysm introduces a phase shift and a smoothing of the pressure profiles, visible on the last row of Figure 7, where the highest flow rates no longer accompany the highest pressures and the largest pressure drops do not exceed 2.8 kPa. The mean pressure drop curve over the observed ICA portion is 0.35 kPa lower in the FSI simulation than in its rigid counterpart, implying a reduced flow resistance.

## 4. Discussion

Simulating wall compliance in IA has been put in question due to the lack of information about tissue properties [34] and minor alleged changes compared to other modelling choices [16]. Voß et al. [15] pointed out the fact that a wrong assumption of the thickness distribution for instance can yield misleading results which jeopardize the use of this more complex simulation type. In our case, while the material properties are unknown, we can use FSI simulations to replicate the observed displacements and assess how the changing domain impacts the hemodynamics. To put our findings into perspective, we are going to evaluate the main rupture hypotheses formulated for delayed aneurysm rupture after FD deployment and comment on how rigid walls tamper with the results.

### 4.1. Pressure Hypothesis

In a widely discussed computational study from 2010, Cebral et al. [8] analyzed the hemodynamics of seven intracranial aneurysms treated with FDs located in the upper ICA. Of these, three ruptured within a week of the intervention, while the remaining four occluded over subsequent months. The healed aneurysms were predominantly medium-sized, except for one giant aneurysm, whereas all ruptured cases belonged to the latter category. The study simulated both preoperative and post-rupture states using similar steps to those employed here, though it did not include compliant wall modelling.

Their findings revealed that, after normalizing inlet pressures to normotensive levels, two of the three ruptured cases exhibited increased intra-aneurysmal pressure post-treatment. The FD deployment had indeed widened stenotic regions upstream of the aneurysm, reducing flow resistance and propagating higher pressures downstream. In the third ruptured case, the stenosis was distal to the aneurysm, and no significant pressure alterations were reported [8].

In our case, the most significant narrowing of the ICA was located at the distal end of the aneurysm, resulting in the observed pressure drop. Assuming that the distal vasculature can compensate for proximal resistance reduction, as described in Cebral’s work, the CFD results do not predict a post-procedural pressure increase. When testing the hypothesis using the FSI modelling, the intrinsic phase shift between the flow rate and the pressure complicates the interpretation of results. In place of the systolic pressure, one may instead consider the time-averaged pressure, which in this case falls by 2.6 mmHg. Whether this modest pressure drop is sufficient to promote the degradation of the aneurysm remains questionable.

Lastly, even when considering the increases in systemic pressure induced by autoregulation, as suggested by Hirato et al. [35], the time scale of pressure-induced ruptures (days) observed in other studies does not align with the delayed rupture in the present case (months).

### 4.2. Thrombosis Hypothesis

Just months after Cebral’s work, Kulcsar et al. [9] proposed an alternative mechanism for the delayed rupture of aneurysms. Their study examined 13 patients with ruptured intracranial aneurysms post-mortem through imaging and autopsies. These patients had giant, saccular, and fusiform aneurysms treated with FDs, with one case involving stent-assisted coiling. Notably, four of the patients reported strong headaches prior to rupture, similar to the present case. The ruptures were classified as early (*n* = 10) or late (*n* = 3), with late ruptures occurring at least three months post-intervention. The latest recorded rupture took place after five months, which is shorter than the eight-month timeline in our case. Additionally, double anti-platelet therapy was reduced to a single agent after three months in Kulcsar’s study, whereas our patient remained on the same regimen throughout.

A common finding in all cases was a significant degree of thrombosis within the aneurysm. Thrombosis is generally considered a positive outcome of FD treatment, as it is often the intended goal. However, in Kulcsar’s study, two dissected aneurysms revealed extensive autolysis and subsequent necrosis of the aneurysm wall [9]. This resulted in pronounced mural thinning, loss of smooth muscle cells, and degradation of fibrous tissue. The authors proposed that thrombosis might disrupt the balance of the aneurysm wall, accelerating rupture. Aneurysm size and aspect ratios (>1.6) were also implicated, as these factors likely facilitated faster thrombus formation.

In the present case of a giant aneurysm, thrombosis was suspected based on hyperdense regions observed on CT imaging (as seen in Figure 8), although no autopsy was performed to confirm it. Thrombosis was first noted one week before rupture and expanded significantly in the interim. These observations align with the disease progression described by Kulcsar et al. [9]. However, the question remains whether thrombosis alone could trigger autolysis rapidly enough to cause rupture within just a few weeks. Kulcsar’s study described a late rupture case where the aneurysm remained fully patent three months post-intervention, with no apparent changes. Twenty days after switching to single anti-platelet therapy, the aneurysm ruptured and was found to be largely thrombosed. If autolysis was indeed responsible for rupture in the cited case, it suggests that the fatal degradation process may require only a short period once initiated. Further studies dedicated to the thrombosis process could provide here valuable information regarding the degradation timeline.

### 4.3. Complex Flow Hypothesis

Motivated by the limited understanding of autolysis and the need to increase cohort sizes for studying delayed rupture, Li et al. [10] conducted numerical investigations on 30 large or giant aneurysms, 10 of which had experienced such an event. Among the evaluated indicators, only one categorical property related to flow stability showed a strong correlation with rupture. This property was described as “flow patterns that persist (do not move or change) during the cardiac cycle”, based on prior work by Cebral et al. [36]. When stable flow patterns were maintained after stenting, the aneurysms were less likely to rupture. The authors proposed that flow structure might influence thrombus formation, potentially linking their findings to the previously discussed thrombus-mediated rupture hypothesis, although they did not provide direct evidence to support this connection.

In the present case, we observed a transition from a “stable” vortex preoperatively to more irregular flow postoperatively in both the CFD and, to a lesser extent, the FSI simulations. This transition aligns qualitatively with observations from other studies [8]. However, determining flow stability remains somewhat subjective, and there is a lack of precise criteria for its assessment. A major limitation of this hypothesis lies in the significant geometric changes following the intervention. In our case, the reshaping of the parent artery predominantly influenced flow dynamics, rather than the presence of the porous mesh at the aneurysm neck. In Li’s study [10], the same geometry was used for both pre-and postoperative simulations, suggesting either that cases like ours were not included or that such geometric changes were overlooked. This discrepancy complicates the direct application of their criteria to our findings.

### 4.4. Limitations

Despite our efforts to achieve high-fidelity modelling of treated aneurysms, certain limitations remain that present opportunities for future refinement.

Firstly, the mechanical parameters of the aneurysm walls were calibrated using observed deformations captured via rotational angiography. This process was inherently limited by the imaging resolution, allowing tracking of only a few discrete points on the aneurysm surface. While our study represents a significant advancement, as it is the first to integrate wall deformations with a realistic deployed stent, there is room for improvement in parameter estimation and tracking methodologies.

Secondly, in the absence of patient-specific data, generalized parameters were employed to define both the inflow waveform and the outflow split, introducing uncertainties. Additionally, thrombus formation in the lower regions of the aneurysm postoperatively caused visible reductions in size. Despite its potential significance, the thrombus was not modelled numerically, as no data regarding its composition were available.

## 5. Conclusions

In this study, we successfully conducted high-fidelity simulations incorporating a fully resolved flow-diverter stent and FSI modelling in a case of delayed aneurysm rupture. Our analysis highlighted significant hemodynamic alterations when wall movements are accounted for, surely due to the low inertia of intra-saccular blood flow in such treated giant aneurysms. Notably, velocity reductions were overestimated in conventional rigid-wall models and pressure profiles were mispredicted, challenging some existing rupture hypotheses.

Among the three examined rupture hypotheses, the thrombosis-mediated failure mechanism emerged as the most plausible explanation in this case. Knowing whether the likelihood of this scenario would be affected by wall movements requires additional work, including the integration of a comprehensive thrombosis model into the current framework. Given that FSI affects intra-aneurysmal flow, it is likely to influence thrombus formation and predictive treatment outcomes. Future studies could offer critical insights into the time scales of thrombus degradation and reveal potential correlations between thrombosis dynamics and the rupture site, enhancing our understanding of post-treatment aneurysm behaviour. 

## Figures and Tables

**Figure 1 bioengineering-12-00305-f001:**
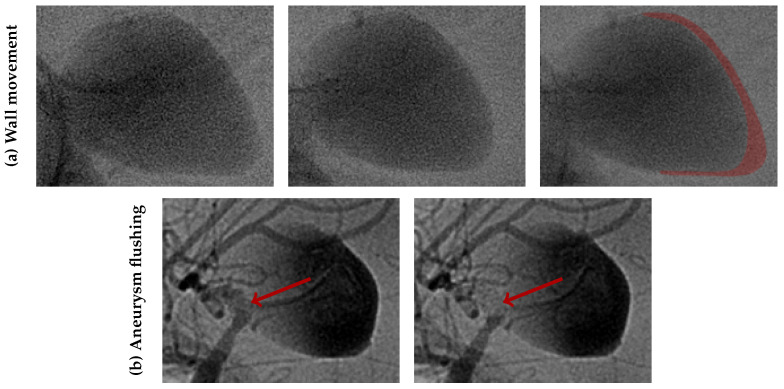
Aneurysm movement on the day of rupture. Wall movement is observed at subsequent snapshots (Δt = 0.17 s) with notable distension of the aneurysm tip (**a**). The last image on the row is an overlay of the previous snapshots with a red-tainted area marking the wall extension. The aneurysm flushing (**b**) is visible during the initial contrast agent injection. During systole, the aneurysm inflates accumulating contrast agent in the distal tip. At diastole, it contracts, pushing blood without contrast agent into the parent vessel (red arrows). The displaced volume is visibly large enough to render portions of the Middle Cerebral artery completely transparent, thereby underscoring the influence of wall compliance.

**Figure 2 bioengineering-12-00305-f002:**
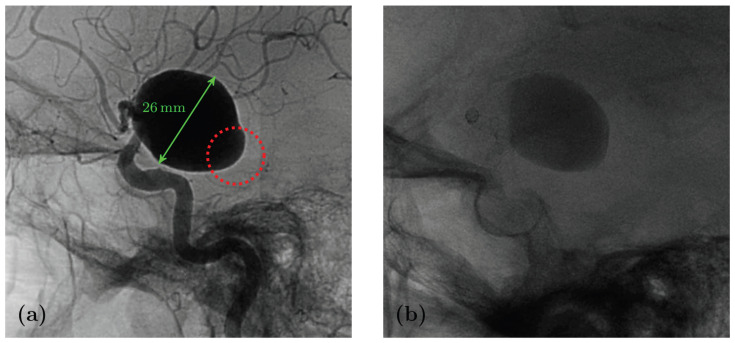
Preoperative (**a**) and post-rupture (**b**) digital subtraction angiography scans. The sac part facing the lateral ventricle where the rupture occurred is circled in red.

**Figure 3 bioengineering-12-00305-f003:**
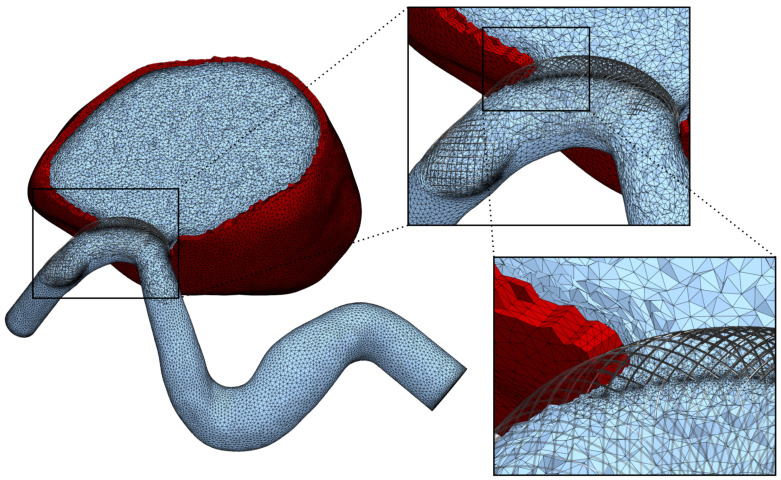
Computational mesh including fluid parts (blue) and the extruded solid mesh (red). The zoomed-in images display in detail the mesh details including isotropic refinement at the stent wires and the unstructured boundary layers.

**Figure 4 bioengineering-12-00305-f004:**
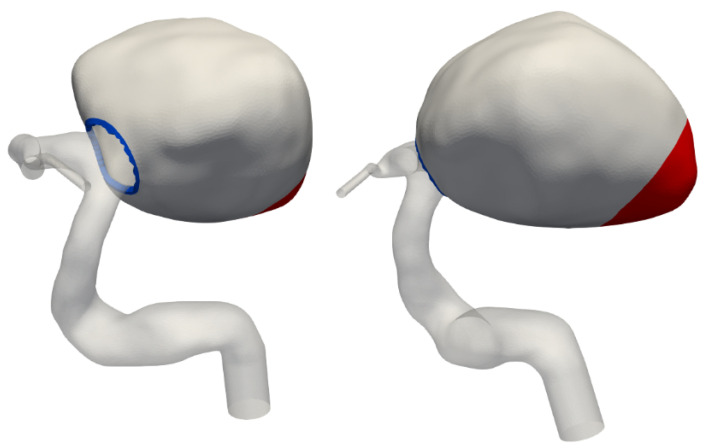
Definition of the weaker aneurysm wall region. The weak spot, highlighted in red, faces the lateral ventricle. The blue rim at the neck is clamped in simulations.

**Figure 5 bioengineering-12-00305-f005:**
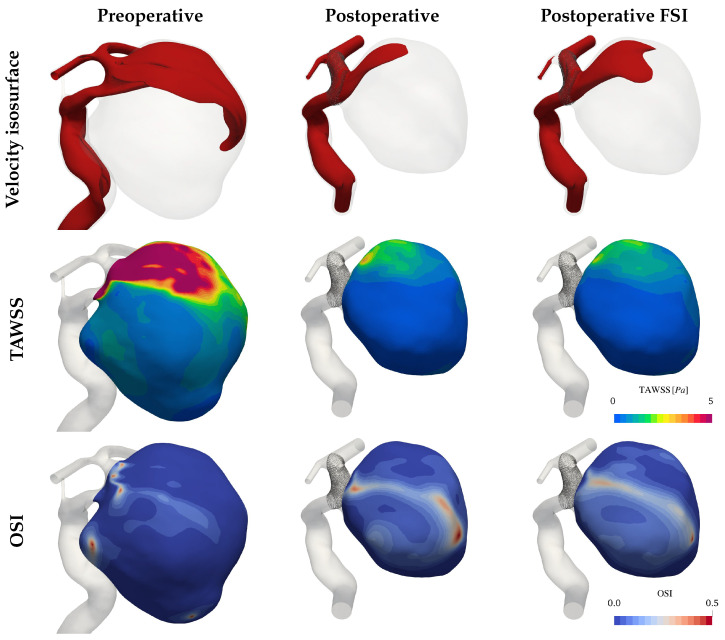
Hemodynamics of the giant aneurysm. The first row shows the isosurface of the flow velocity 0.4 m/s at systole (t = 0.11 s). The second row shows the Time-Averaged WSS (TAWSS) on the bulge, and the last indicates the Oscillatory Shear Index (OSI) distributions.

**Figure 6 bioengineering-12-00305-f006:**
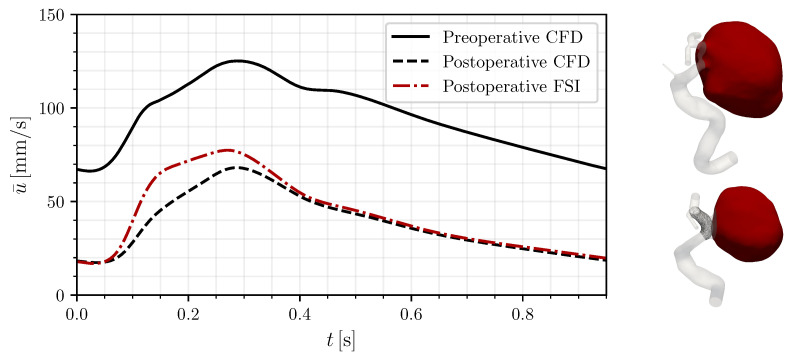
Averaged bulge velocity $\bar{u}$ of the pre- and postoperative cases including FSI. On the right, the areas considered for the computation are overlayed in red for both cases.

**Figure 7 bioengineering-12-00305-f007:**
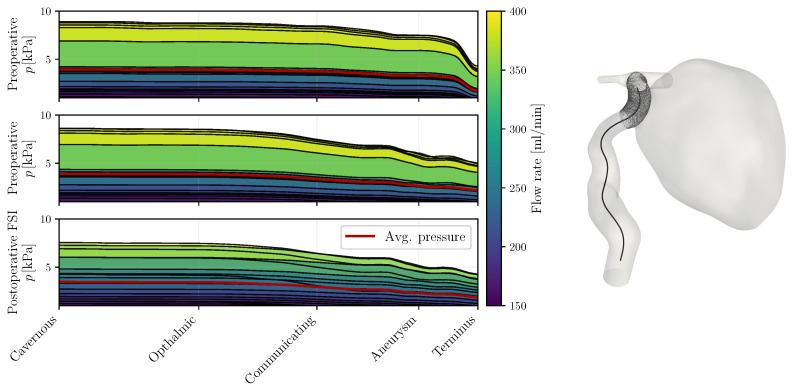
Pressure along the centerline recorded in the preoperative and postoperative cases. The centerline is shown on the right for the postoperative geometry. For the latter, the results are given for the CFD and FSI computations. The pressure curves have been levelled at the minimal terminal ICA pressure for reference. The time-averaged pressure curve is inscribed in red for each simulation.

**Figure 8 bioengineering-12-00305-f008:**
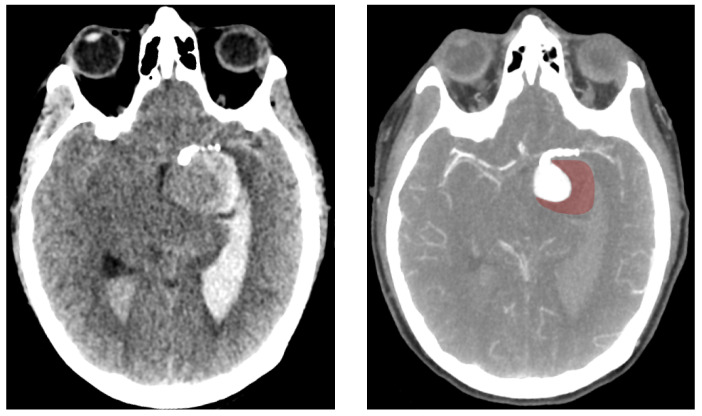
Computed Tomography (CT) scan of the ruptured aneurysm (**left**) showing the presence of blood in both ventricles. On the (**right**), a CT scan using a contrast agent which highlights the arteries and the perfused sections of the aneurysm. The red-tainted section corresponds to the thombosed regions at which the rupture occurred.

## Data Availability

Parts of the data can be shared upon request, subject to prior contact with the authors and compliance with ethical and privacy regulations.

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
