# Peer review of "Investigating Delayed Rupture of Flow Diverter-Treated Giant Aneurysm Using Simulated Fluid–Structure Interactions"

_bioengineering, 2025, doi:10.3390/bioengineering12030305_

Round 1

Reviewer 1 Report

Comments and Suggestions for Authors

General Comments
These authors performed an image-based computational study of a giant intracranial aneurysm case treated with flow diverter. The study presents pre-operative computational fluid dynamics (CFD) and then post-operative CFD together with a fluid-structure interaction (FSI) simulation, of which resulting hemodynamics are compared to established delayed-rupture hypotheses. The authors concluded that, when considering walls compliance as compared to rigid ones, there is a phase shift between the flow rate and the pressure, dampening in pressure cycles, and increased aneurysmal flow. 

The article is well-written, and the authors are to be commended for undertaking such a study.

Minor: Page 5, line 152: Fig. 6 should be Fig.5. 

Author Response

We thank the reviewer for employing his time in the review and for spotting the reference mistake.

Reviewer 2 Report

Comments and Suggestions for Authors
  • The title could be more specific regarding "Simulated Fluid-Structure Interactions" to better inform readers about the modeling approach used.

  • In the abstract, the phrase "The resulting hemodynamics are systematically compared to established delayed-rupture hypotheses" could be clearer by briefly stating which hypotheses are examined.

  • The introduction mentions a 20-fold higher rupture risk but lacks a direct citation at that point. Ensure that this claim is properly referenced.

  • In Figure 1, the red arrow is described in the text but is not very prominent. Consider adding a clearer annotation to enhance readability.

  • The mesh resolution in Section 2.2 is described, but further justification for selecting the 0.25–0.3 mm element size could strengthen the methodology.

  • Some equations in Section 2.3, particularly Equation A1, could be formatted more consistently, as the subscripts could be better aligned.

  • In the discussion, the thrombosis hypothesis section could briefly mention how computational modeling could further validate or challenge this mechanism, as it is largely based on clinical observations.

  • The limitations section acknowledges resolution limitations in tracking aneurysm wall deformations, but it might help to mention whether higher-resolution imaging techniques, such as 4D flow MRI, could improve future studies.

Author Response

We thank reviewer 2 for the insightful comments which have notably helped improving the quality of the work. Attached, we send a word document with the answers and comments to each of the remarks.

Reviewer 3 Report

Comments and Suggestions for Authors

I thank the editor for the opportunity to review the manuscript.  

The study compares various models for assessing flow simulation and the resulting pressure gradients in aortic aneurysms. Furthermore, it evaluates potential factors influencing the rupture of this structure, which can have catastrophic consequences. The manuscript is well-written and clearly structured.  

The authors accurately conclude that further studies are necessary for the implementation of the proposed methodology. As such, this work serves as an introduction to a more comprehensive future study.  

I recommend accepting the manuscript in its current form.

Author Response

Thank you for the time employed on the review.